# How noise affects the Hessian spectrum in overparameterized neural networks

## Abstract

Stochastic gradient descent (SGD) forms the core optimization method for deep neural networks. While some theoretical progress has been made, it still remains unclear why SGD leads the learning dynamics in overparameterized networks to solutions that generalize well. Here we show that for overparameterized networks with a degenerate valley in their loss landscape, SGD on average decreases the trace of the Hessian of the loss. We also generalize this result to other noise structures and show that isotropic noise in the non-degenerate subspace of the Hessian decreases its determinant. In addition to explaining SGDs role in sculpting the Hessian spectrum, this opens the door to new optimization approaches that may confer better generalization performance. We test our results with experiments on toy models and deep neural networks.

## 1 Introduction

Deep neural networks have achieved remarkable success in the past decade on tasks that were out of reach prior to the era of deep learning. Yet fundamental questions remain regarding the strong performance of over-parameterized models and optimization schemes that typically involve only first-order information, such as stochastic gradient descent (SGD) and its variants.

Regarding generalization, it has been noted that flat minima with small curvature tend to generalize better than sharp minima (Hochreiter & Schmidhuber, 1997; Keskar et al., 2017). This has been argued by nonvacuous PAC-Bayes bounds (Dziugaite & Roy, 2017) and Bayesian evidence (Smith & Le, 2018). But why does SGD bias learning towards flat minima? One possible explanation was proposed by Zhang et al. (2018) in terms of energy-entropy competition. Jastrzbski et al. (2018) also suggests that isotropic noise in SGD helps networks escape sharp minima with large Hessian determinant. However, previous theoretical analyses assume that minima are isolated and non-singular. In contrast, Sagun et al. (2017b) finds that most of the eigenvalues of the Hessian of the loss function at a minimum are close to zero, indicating highly degenerate minima. The degeneracy is further supported by results on mode connectedness (Garipov et al., 2018; Draxler et al., 2018). Furthermore, it is shown that minima found from different initializations and the same optimization scheme are connected with essentially no barriers between them. Nguyen (2019) also shows theoretically that for a class of deep overparameterized neural nets with piecewise linear activation functions, all of the global minima are connected within a unique and potentially very large global valley.

In this paper, we prove that for models whose loss has a "minimal valley" structure, defined below, optimization via SGD decreases the trace of Hessian of the loss, by utilizing recent fluctuation-dissipation relations (Yaida, 2019). Furthermore, we derive the noise covariance matrix that would result in the reduction of other potentially desired quantities such as the Hessian determinant, leading towards the design of new optimization algorithms. We present experiments on toy models and deep neural networks to confirm our predictions.

## 2 Main Theorem

In this section, we present our main theorem - how noise during optimization affects the Hessian of the loss function when the loss landscape locally takes the shape of a degenerate valley. More specifically, let $N$ and $n$ be the dimension of the total parameter space and non-degenerate space, respectively, and

consider a general loss function $L(\boldsymbol{w})$ where $\boldsymbol{w} \in \mathbb{R}^N$ are the model parameters.[1] Since the curvature around the minima can vary, we approximate the loss function in such a valley around a degenerate minimum with a modified quadratic form $L(\boldsymbol{w}) = L^* + \frac{1}{2}(\boldsymbol{w} - \mathcal{P}(\boldsymbol{w}))^{\mathrm{T}} H(\mathcal{P}(\boldsymbol{w}))(\boldsymbol{w} - \mathcal{P}(\boldsymbol{w}))$ where $\mathcal{P}$ is a function that projects a point $w$ in parameter space to the nearest minimum, and $H$ is the Hessian, a function of the location of the projected minimum. We have the following lemma:

**Lemma 1.** *For an arbitrary point $\boldsymbol{w}$ and its neighborhood in the valley, there exists an orthogonal transformation $Q$ and a translation vector $\boldsymbol{v}$ such that the loss function in the new coordinate system $\boldsymbol{\theta} = Q\boldsymbol{w} - \boldsymbol{v}$ has the following form,*

$$L(\boldsymbol{\theta}) = L^* + \frac{1}{2} \sum_{i=1}^{n} \theta_i^2 \lambda_i(\theta_{n+1}, ..., \theta_N) \ , \tag{1}$$

*where $\lambda_i$s are the positive eigenvalues of the loss function Hessian for the non-degenerate space and depend on the position in the degenerate space. Also, note that the gradient descent equation is invariant under this transformation.*

A detailed, constructive proof of this lemma can be found in Appendix A.1. Notice that this is a generalization of Morse's Lemma (Callahan, 2010). In the rest of this section, we denote the nondegenerate and degenerate subspaces by $\bar{\boldsymbol{\theta}} = (\theta_1, ..., \theta_n)^{\mathrm{T}}$ and $\hat{\boldsymbol{\theta}} = (\theta_{n+1}, ..., \theta_N)^{\mathrm{T}}$, respectively. Similarly, the gradients of $\bar{\boldsymbol{\theta}}$ and $\hat{\boldsymbol{\theta}}$ are denoted by $\bar{\nabla}$ and $\hat{\nabla}$, respectively. Notice that at a minimum where $\bar{\boldsymbol{\theta}} = 0$, the $\lambda_i$s are the only nonzero Hessian eigenvalues.

Next we provide a quick review of the relevant fluctuation-dissipation relation formalism for stochastic gradient descent (Yaida, 2019). We denote the loss for a random batch $B$ of training sample as $L^B(\boldsymbol{\theta})$. Clearly we have

$$[\![\nabla L^B(\boldsymbol{\theta})]\!]_{\mathrm{m.b.}} = \nabla L(\boldsymbol{\theta}) \ , \tag{2}$$

where $[\![\bullet]\!]_{\mathrm{m.b.}}$ represents the average over all mini-batch realizations. If there exists a stationary state for $\bar{\boldsymbol{\theta}}$, $p_{ss}(\bar{\boldsymbol{\theta}})$ and the stationary-average is defined as $\langle \mathcal{O}(\bar{\boldsymbol{\theta}}) \rangle \equiv \int d\bar{\boldsymbol{\theta}} p_{ss}(\bar{\boldsymbol{\theta}}) \mathcal{O}(\bar{\boldsymbol{\theta}})$ where $\mathcal{O}(\bar{\boldsymbol{\theta}})$ is any observable of $\bar{\boldsymbol{\theta}}$, it is straightforward to derive the following master equation:

$$\langle \mathcal{O}(\bar{\boldsymbol{\theta}}) \rangle = \langle [\![\mathcal{O}[\bar{\boldsymbol{\theta}} - \eta \bar{\nabla} L^B(\boldsymbol{\theta})]]\!]_{\mathrm{m.b.}} \rangle \ . \tag{3}$$

We denote the two-point noise matrix as $\tilde{C}_{i,j}(\theta) \equiv [\![\partial_{\theta_i} L^B(\boldsymbol{\theta}) \partial_{\theta_j} L^B(\boldsymbol{\theta})]\!]_{\mathrm{m.b.}}$ and the noise covariance matrix $C_{i,j}(\theta) \equiv \tilde{C}_{i,j}(\theta) - \partial_{\theta_i} L(\boldsymbol{\theta}) \partial_{\theta_j} L(\boldsymbol{\theta})$.

Now we present our main theorem:

**Theorem 1.** *When the loss can be locally approximated as in Equation 1, assuming that the non-degenerate space $\bar{\boldsymbol{\theta}}$ is in a stationary state at time $t$ and that the noise covariance matrix is aligned with the Hessian, we have*

$$\langle [\![T_{\mathrm{t+1}} - T_{\mathrm{t}}]\!]_{\mathrm{m.b.}} \rangle \leq 0 \ ,$$

*where $T = \sum_{i=1}^{n} \lambda_i$ is the trace of the Hessian and $T_{\mathrm{t}}$ represents the trace at training step $t$. Equality holds when $\nabla L^B(\boldsymbol{\theta}) = \nabla L(\boldsymbol{\theta})$ or $\hat{\nabla} T(\hat{\boldsymbol{\theta}}) = 0$.*

Theorem 1 indicates that the change of the Hessian trace during SGD optimization is non-positive on average. A trivial condition for equality is optimization with gradient descent (GD) instead of its stochastic variant because the stationary state for noiseless optimization is a constant state with vanishing gradient. An alternate condition for equality is that the trace function of $\bar{\boldsymbol{\theta}}$ reaches a minimum or saddle point.

Before proving Theorem 1, we will first analyze the assumptions made. We emphasize at the outset that we expect our assumptions to be valid only after an early phase of training corresponding to $\sim 10$ epochs in our experiments.

## 2.1 EVIDENCE FOR THE ASSUMPTIONS MADE

In this subsection, we will analyze the assumptions in Theorem 1 and present some experiments that support their validity.

---

[1] Here the non-degenerate space corresponds to the subspace of the Hessian with non-zero eigenvalues. Note that we do *not* require any specific relationship between $n$ and $N$, e.g. $n \ll N$.

### 2.1.1 MINIMAL VALLEY

Our theory relies on the existence of a degenerate valley in the loss function defined previously via Equation 1. In addition to degeneracy, we furthermore assume that there exists a region of connected degenerate minima in the landscape of the loss function, in particular for overparameterized models. SGD or its variants then selects one of the degenerate solutions. Such degeneracy has been observed empirically where most of the eigenvalues of the loss Hessian from various models and tasks are zero or close to zero (Sagun et al., 2017a;b). Furthermore, the connectedness is further supported by results of mode connectedness (Garipov et al., 2018; Draxler et al., 2018). Nguyen (2019) also shows theoretically that for a class of deep overparameterized neural nets with piecewise linear activation functions, all of the global minima are connected within a unique and potentially very large global valley.

Here we present additional experiments to support the existence of such *minimal valleys*, i.e. basins of attraction of connected, degenerate minima that can be locally described by Equation 1. These experiments will be also used to analyze and confirm our theory later. We consider classification of CIFAR10 with label-smoothing cross entropy (Szegedy et al., 2016)[2], constant learning rate of $0.1$, momentum $0.9$, batch size $500$ and total training epochs $150$. The network architectures are Preact-ResNet18 (He et al., 2016) and VGG16 (Simonyan & Zisserman, 2015). The training and validation loss are shown in Figure 1 (inset). After $150$ training epochs, the networks fluctuates with small training loss, and the minimal training loss for both case is $\sim 0.005$.

In order to confirm the existence of minimal valleys, we would like to project each point along the training path to the corresponding closest minimum [3] and check whether the projected path crosses any barriers. To determine whether two minima can be connected by a line segment in a minimal valley implying no barriers in between, we use line interpolation to connect these two minima and measure the training loss along the interpolating line. In practice, we call it line connectedness if we sample evenly $10$ points in the line segment between consecutive minima and the training loss is smaller than a threshold for all $10$ points.

To project a state to a closest minimum, the model is trained from that state with the same hyperparameters except using GD instead of SGD to remove noise. The stopping criterion for these auxiliary training tasks is that the training loss is less than the minimal training loss, which is $\sim 0.005$.

Ideally, we would project states after each iteration and check for line connectedness, but this involves large unnecessary calculations. In practice, if state B is obtained from $n$ training iterations after state A, and A and B are not line connected, we insert C which is obtained from $n/2$ training iterations after A and check for line connectedness between A and C and between C and B. This process stops when there is no extra iteration between A and B or they are line connected. Starting from each pair of consecutive epochs as A and B, we obtain the projected path recursively. The training and validation losses on these two projected paths are shown in Figure 1 (main plot). The left path has $2700$ projected points and the right has $4020$. We see that after a few epochs, the training loss along the path remains close to zero (and hence minimal), which means that there exists a least one minimal valley connected to the model state found by regular SGD. Also SGD optimization happens within such a valley after the first few epochs. Furthermore, the validation loss varies along the valley.

More experiments can be found in Appendix A.2. Further discussion of violation of this minimal valley assumption can be found in Appendix A.6.

### 2.1.2 THE HESSIAN-NOISE COVARIANCE ALIGNMENT

Empirical observations have found that the noise covariance matrix is aligned with the Hessian (Zhang et al., 2018; Zhu et al., 2019; Jastrzbski et al., 2018). In fact, it was shown in Hoffer et al. (2017) that

$$C_{i,j} = \frac{1}{S}\left(1 - \frac{S}{M}\right)\frac{1}{M}\sum_{k=1}^{M}\partial_{\theta_i}l(x_k, \boldsymbol{\theta})\partial_{\theta_j}l(x_k, \boldsymbol{\theta})\ ,$$

---

[2]The label-smoothing technique is used here to make a true minimum of the cost function exist. Regular cross entropy gives similar results.

[3]We use minimum here but specifically we mean a point with training loss less than a threshold.

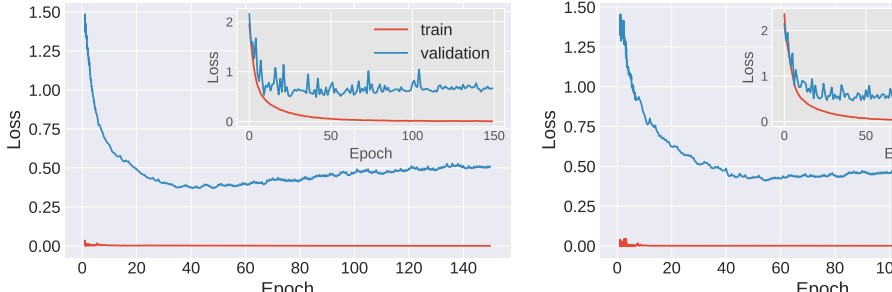

Figure 1: CIFAR10 trained on Preact-ResNet18 (left) and VGG16 (right). Main plots show training and validation loss of projected paths. Projected paths are obtained by projecting each model state during optimization to a corresponding state found by GD with small training loss. Each pair of consecutive projected states are then connected by line interpolation. The inset shows training and validation loss of training paths.

where $S$ is the mini-batch size and $M$ is the total number of training samples. Considering negative log-likelihood loss functions, which is also the loss function in our experiments for classification, the loss $L$ can be written as $L = -\frac{1}{M} \sum_{i=k}^{M} \log f(x_k, \boldsymbol{\theta})$ where $f$ is the output of the network and $l(x_k, \boldsymbol{\theta}) = -\log f(x_k, \boldsymbol{\theta})$. Notice that

$$\partial_{\theta_i} \partial_{\theta_j} L(\boldsymbol{\theta}) = \frac{1}{M} \sum_{k=1}^{M} \partial_{\theta_i} l(x_k, \boldsymbol{\theta}) \partial_{\theta_j} l(x_k, \boldsymbol{\theta}) - \frac{1}{M} \sum_{k=1}^{M} \frac{1}{f(x_k, \boldsymbol{\theta})} \partial_{\theta_i} \partial_{\theta_j} f(x_k, \boldsymbol{\theta}) \ , \tag{4}$$

where the left-hand side is the positive Hessian matrix for $i, j \leq n$ and the first term on the right-hand side is the positive matrix proportional to the noise covariance matrix. Empirically, negative eigenvalues are exponentially small in magnitude compared to major positive ones when a model is close to a minimum (Sagun et al., 2017b) and negative eigenvalues can only come from the contribution of the second term on the right-hand side of equation (4). Therefore unless the second term is extremely asymmetrical, we can assume that its contribution is small compared to the first term. We will return to this in Section 4.1 and show that eigenvalues with small magnitude arise from higher-order curvature in the degenerate space even when the Hessian with respect to the bottom of the valley is characterized by the positive $\lambda_i$s for $i = 1, ..., n$ or zero directions. Ignoring the second term on the right hand side, Equation 4 becomes

$$C_{i,j} = \frac{1}{S} \left( 1 - \frac{S}{M} \right) H_{i,j} \ . \tag{5}$$

We emphasize that this assumption will only be valid after the early phase of training, not from initialization. Further discussion and experimental support can be found in Appendix A.4.

### 2.1.3 TIMESCALE SEPARATION

The final assumption is that the dynamics relaxes to a stationary state in the non-degenerate space $\bar{\boldsymbol{\theta}}$ but not in the degenerate space. This assumption is because there is a timescale separation between the dynamics of $\bar{\boldsymbol{\theta}}$, which relax quickly and the dynamics of $\hat{\boldsymbol{\theta}}$, which evolve much more slowly as the minimal valley is traversed.

Considering a simplified optimization problem where parameter directions undergo independent Levy-driven Ornstein-Uhlenbeck processes[4] as for quadratic loss with noise, the decay to a stationary state is exponential with decay rate proportional the corresponding eigenvalues (Abdelrazeq et al., 2014). Since the $\bar{\boldsymbol{\theta}}$ correspond to the leading order eigenvalues, their relaxation time is exponentially shorter than the evolution in the degenerate space. Therefore it is reasonable to assume a stationary state in the non-degenerate space $\bar{\boldsymbol{\theta}}$ when studying the degenerate $\hat{\boldsymbol{\theta}}$ dynamics. Further discussion and experimental support of this assumption using fluctuation-dissipation relations can be found in Appendix A.5.

---

[4]Simsekli et al. (2019) proposes to analyze SGD as an SDE driven by a Levy motion.

## 2.2 Proof of Theorem 1

*Proof.* The stochastic gradient descent update for $\hat{\boldsymbol{\theta}}$ is $\hat{\boldsymbol{\theta}}_{t+1} = \hat{\boldsymbol{\theta}}_t - \eta\hat{\nabla}L^B(\boldsymbol{\theta}_t)$. If we consider the average evolution of the eigenvalues $\lambda_i$ for $i = 1, ..., n$ from step t to t + 1, we have

$$
\begin{aligned}
\langle[\![\lambda_{i,t+1} - \lambda_{i,t}]\!]_{\text{m.b.}}\rangle &= \langle[\![\lambda_i(\hat{\boldsymbol{\theta}}_{t+1}) - \lambda_i(\hat{\boldsymbol{\theta}}_t)]\!]_{\text{m.b.}}\rangle \\
&= \langle[\![\lambda_i(\hat{\boldsymbol{\theta}}_t - \eta\hat{\nabla}L^B(\boldsymbol{\theta}_t)) - \lambda_i(\hat{\boldsymbol{\theta}}_t)]\!]_{\text{m.b.}}\rangle \\
&= -\eta\langle[\![\hat{\nabla}\lambda_i(\hat{\boldsymbol{\theta}}_t)^{\text{T}}\hat{\nabla}L^B(\boldsymbol{\theta}_t))]\!]_{\text{m.b.}}\rangle + O(\eta^2) \\
&= -\eta\langle\hat{\nabla}\lambda_i(\hat{\boldsymbol{\theta}}_t)^{\text{T}}\hat{\nabla}L(\boldsymbol{\theta}_t))\rangle + O(\eta^2) \ ,
\end{aligned}
\tag{6}
$$

where we performed a Taylor expansion from the second to the third equality and the fourth equality holds from equation (2).

Notice that $\hat{\nabla}L(\boldsymbol{\theta}) = \frac{1}{2}\sum_{i=1}^n \theta_i^2\hat{\nabla}\lambda_i(\hat{\boldsymbol{\theta}})$. By keeping the leading order term, we have

$$
\langle[\![\lambda_{i,t+1} - \lambda_{i,t}]\!]_{\text{m.b.}}\rangle \simeq -\frac{\eta}{2}\sum_{j=1}^n \langle\theta_j^2\rangle\hat{\nabla}\lambda_i(\hat{\boldsymbol{\theta}}_t)^{\text{T}}\hat{\nabla}\lambda_j(\hat{\boldsymbol{\theta}}_t) \ .
\tag{7}
$$

Considering the observables $\mathcal{O}(\bar{\boldsymbol{\theta}}) \equiv \theta_i^2$ for $i = 1, ..., n$, the master equation (3) becomes

$$
\langle\theta_i\partial_{\theta_i}L\rangle = \frac{\eta}{2}\langle\tilde{C}_{i,i}\rangle \ .
$$

Notice that $\partial_{\theta_i}L = \theta_i\lambda_i(\hat{\boldsymbol{\theta}})$, we thus have

$$
\langle\theta_i^2\rangle = \frac{\eta\langle\tilde{C}_{i,i}\rangle}{2\lambda_i(\hat{\boldsymbol{\theta}})} \ .
\tag{8}
$$

By the definition of the two-point noise matrix $\tilde{C}$ and the noise covariance matrix $C$, we also have

$$
\begin{aligned}
\tilde{C}_{i,i} &= C_{i,i} + \partial_{\theta_i}L(\boldsymbol{\theta})\partial_{\theta_i}L(\boldsymbol{\theta}) \\
&= C_{i,i} + \theta_i^2\lambda_i^2(\hat{\boldsymbol{\theta}})
\end{aligned}
\tag{9}
$$

Based on the assumption that the noise covariance matrix is aligned with the Hessian and scales as $\frac{1}{S}\left(1 - \frac{S}{M}\right)$ where $S$ is the mini-batch size and $M$ is the total number of training samples (Hoffer et al., 2017)[5], we have

$$
C_{i,i} = \frac{1}{S}\left(1 - \frac{S}{M}\right)\lambda_i(\hat{\boldsymbol{\theta}}) \ .
\tag{10}
$$

Together with Eq. 8, 9 and 10, we have for $i \le n$,

$$
\begin{aligned}
\langle\theta_i^2\rangle &= \frac{\eta}{S(2 - \eta\lambda_i)}\left(1 - \frac{S}{M}\right) \\
&= \frac{\eta}{2S}\left(1 - \frac{S}{M}\right) + O(\eta^2)
\end{aligned}
\tag{11}
$$

Now Eq. 7 becomes

$$
\langle[\![\lambda_{i,t+1} - \lambda_{i,t}]\!]_{\text{m.b.}}\rangle = -\frac{\eta^2}{4S}\left(1 - \frac{S}{M}\right)\sum_{j=1}^n \hat{\nabla}\lambda_i(\hat{\boldsymbol{\theta}}_t)^{\text{T}}\hat{\nabla}\lambda_j(\hat{\boldsymbol{\theta}}_t) + O(\eta^3)
$$

Next we consider the evolution of the trace of the Hessian $T = \sum_{i=1}^n \lambda_i$. We have

$$
\begin{aligned}
\langle[\![T_{t+1} - T_t]\!]_{\text{m.b.}}\rangle &= -\frac{\eta^2}{4S}\left(1 - \frac{S}{M}\right)\sum_{i,j=1}^n \hat{\nabla}\lambda_i(\hat{\boldsymbol{\theta}}_t)^{\text{T}}\hat{\nabla}\lambda_j(\hat{\boldsymbol{\theta}}_t) + O(\eta^3) \\
&\approx -\frac{\eta^2}{4S}\left(1 - \frac{S}{M}\right)\left\|\sum_{i=1}^n \hat{\nabla}\lambda_i(\hat{\boldsymbol{\theta}})\right\|^2 \\
&\le 0 \ .
\end{aligned}
$$

---

[5]Refer to Assumption 2.1.2 for details.

Thus we have shown that the trace of Hessian decreases on average during SGD optimization. Notice that equality holds when either one of the following conditions is satisfied: first, if $S = M$ so that SGD becomes full-batch GD, and second that $\sum_{i=1}^{n} \hat{\nabla}\lambda_i(\hat{\boldsymbol{\theta}}) = 0$, which implies $\hat{\nabla}T(\hat{\boldsymbol{\theta}}) = 0$.

$\square$

## 3 OPTIMIZATION DYNAMICS WITH NOISE BEYOND SGD

In Section 2, we concluded that, to leading order in the learning rate $\eta$, SGD dynamics reduces the Hessian trace. This result originates from the noise introduced by SGD with covariance given by (10). Interestingly, other forms of noise can be introduced into gradient descent in order to decrease other desired quantities instead of the trace. Here we present a theorem that relates the expected decrease of certain functions of the Hessian spectrum to a corresponding noise covariance. And $\langle \bullet \rangle$ represents an average over the stationary state and $[\![\bullet]\!]$ averages the quantity over the corresponding noise.

**Theorem 2.** *In a minimal valley with loss approximation in Equation 1, assuming that there exists a stationary state in the non-degenerate space $\bar{\boldsymbol{\theta}}$, for any quantity $f(\boldsymbol{\lambda})$ satisfying $\frac{\partial f(\boldsymbol{\lambda})}{\partial \lambda_i} \geq 0$ for $i = 1, ..., n$, where $\boldsymbol{\lambda} \equiv (\lambda_1, ..., \lambda_n)^{\mathrm{T}}$, if the loss is optimized with gradient descent along with external noise with diagonal covariance matrix*

$$C_{i,i} = \lambda_i \frac{\partial f(\boldsymbol{\lambda})}{\partial \lambda_i} \ , \tag{12}$$

*we have*

$$\langle [\![ f_{t+1} - f_t ]\!] \rangle \leq 0 \ ,$$

where $f_t$ represents $f(\boldsymbol{\lambda})$ evaluated at training step $t$. Equality holds when $\hat{\nabla}f(\boldsymbol{\lambda}(\hat{\boldsymbol{\theta}})) = 0$.

*Proof.* Similar to the proof of Theorem 1, we have

$$\langle [\![ f_{t+1} - f_t ]\!] \rangle = -\frac{\eta}{2} \sum_{i=1}^{n} \frac{\partial f}{\partial \lambda_i} \sum_{j=1}^{n} \langle \theta_j^2 \rangle \hat{\nabla}\lambda_i^{\mathrm{T}} \hat{\nabla}\lambda_j + O(\eta^2) \ . \tag{13}$$

With the imposed noise with covariance matrix (12), the master equation (3) and the relation between noise covariance matrix and two-point noise matrix in equation (9), we have

$$
\begin{aligned}
\langle \theta_i^2 \rangle &= \frac{\eta}{(2 - \eta\lambda_i)} \frac{\partial f}{\partial \lambda_i} \\
&= \frac{\eta}{2} \frac{\partial f}{\partial \lambda_i} + O(\eta^2) \ .
\end{aligned}
\tag{14}
$$

Plugging in Equation 13 we have

$$
\begin{aligned}
\langle [\![ f_{t+1} - f_t ]\!] \rangle &= -\frac{\eta^2}{4} \Big\| \sum_{i=1}^{n} \frac{\partial f}{\partial \lambda_i} \hat{\nabla}\lambda_i(\hat{\boldsymbol{\theta}}) \Big\|^2 + O(\eta^3) \\
&\approx -\frac{\eta^2}{4} \Big\| \sum_{i=1}^{n} \hat{\nabla}f(\boldsymbol{\lambda}(\hat{\boldsymbol{\theta}})) \Big\|^2 \\
&\leq 0 \ .
\end{aligned}
\tag{15}
$$

Thus we have shown that $f$ decreases on average during optimization. Equality holds when $\hat{\nabla}f(\boldsymbol{\lambda}(\hat{\boldsymbol{\theta}})) = 0$. Notice that Theorem 1 is a special case with $f(\boldsymbol{\lambda}) = \sum_{i=1}^{n} \lambda_i$. $\square$

Furthermore, we have the following corollary:

**Corollary 2.1** (Determinant-Decreasing Noise). *With the same assumptions in Theorem 2, let $f(\boldsymbol{\lambda}) = \log \prod_{i=1}^{n} \lambda_i$ and $C_{i,i} = C$ where $C$ is a arbitrary positive constant, we have*

$$\langle [\![ \mathrm{Det}_{t+1} - \mathrm{Det}_t ]\!] \rangle \leq 0 \ ,$$

*where $\mathrm{Det} \equiv \prod_{i=1}^{n} \lambda_i$ is the non-degenerate determinant.*

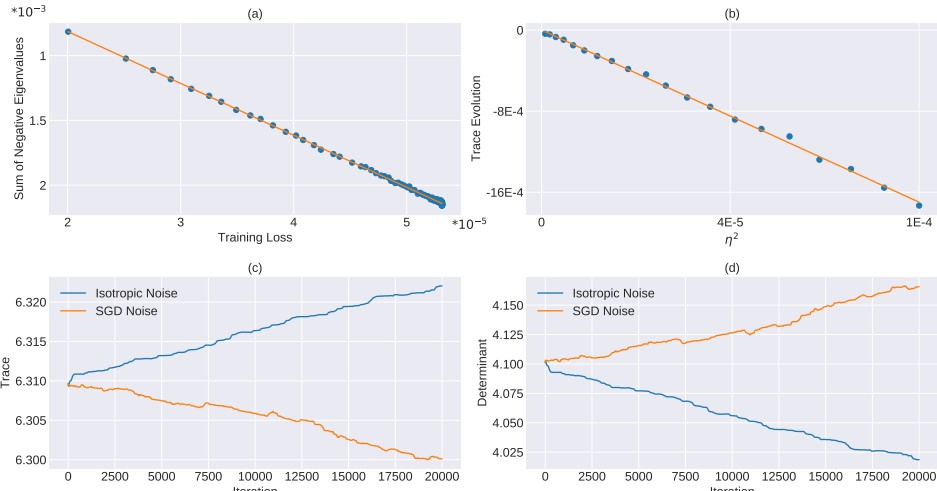

Figure 2: (a) The training loss v.s. the sum of negative eigenvalues. Red is the best fit line, $y = wx + b$ with $w = -40.07$ and $b = -1.22 * 10^{-5}$. The data is nearly a straight line and the y-intercept is almost zero, consistent with our prediction in Appendix A.3. (b) The trace evolution v.s. the squared learning rate. The result shows a linear relation. Red is best fit line. (c)-(d) Change of the Hessian trace and determinant during training using isotropic (blue) and SGD noise (red).

Corollary 2.1 indicates that the non-degenerate determinant will decrease on average during optimization if we introduce isotropic noise in the non-degenerate space. It is known (Smith & Le, 2018) that the Bayesian evidence, minimization of which minimizes a PAC-Bayes generalization bound, has a contribution from the non-degenerate determinant. This contribution is also called the Occam factor. Our result thus leads to a new algorithm where one introduces isotropic noise into the non-degenerate space. Theorem 2 will allow future work to add designed noise that applies an entropic force on other properties of the Hessian yet to be determined.

## 4    EXTENDED ANALYSIS AND EXPERIMENTS

### 4.1    NEGATIVE EIGENVALUES OF THE HESSIAN

Empirically, small negative eigenvalues arise even with small training loss, which seems to indicate that the model is at a saddle point instead of a minimum (Sagun et al., 2017b; Ghorbani et al., 2019). This however is consistent with the loss in Equation 1. The loss is minimal when $\theta_i = 0$ for $i \leq n$. However, when we use a trained instance to estimate the eigenvalues of the Hessian, we only guarantee that $\theta_i$ is close to zero (and thus nonzero training loss) for $i \leq n$. Therefore, there could exist negative eigenvalues which originate from the second derivative of $\theta_i$ for $i > n$. They are originally zero at the minimum with $\theta_i = 0$ for $i \leq n$, and their magnitude is suppressed by $\theta_i^2$ for $i \leq n$ which is related to the small training loss.

In Appendix A.3, we predict that the negative eigenvalues on average are proportional to the training loss. To confirm this, we set up a simple experiment to classify MNIST data with a small training set of 500 randomly selected samples, with a single hidden layer fully-connected network with 6220 parameters and loss given by label-smoothed cross entropy, again to make a genuine minimum. We first obtain a minimum with gradient descent optimization at learning rate 0.1 and $10k$ training epochs and use this as initialization for later. The model is then further trained from this initialization with different batch sizes from 5 to 250 and learning rates from 0.001 to 0.1 in order to equilibrate to various training losses. Then the negative eigenvalues are calculated and the result is shown in Fig. 2(a). The result shows a linear relationship between training loss and the sum of negative eigenvalues, as predicted.

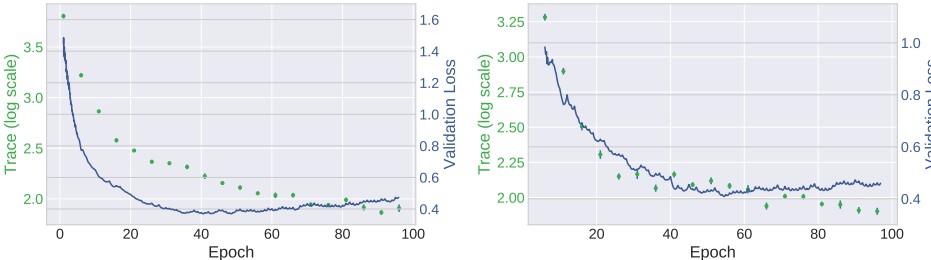

Figure 3: Hessian Trace estimation of projected paths found in Section 2 with network architecture of Preact-ResNet18 (left) and VGG16 (right). Red bars represent the errors in trace estimation. The decreasing trace during SGD optimization confirms our theory prediction.

## 4.2 The Evolution of Trace and Determinant with a Toy Model

In this section, we use two toy models to test our previous theoretical analysis. In Section 2, we showed that the trace evolution rate is proportional to $\eta^2$. Notice that the two factors of $\eta$ have different origins. One is from the expansion of $\lambda$ and the gradient descent step as in Eq. 6. The other contribution is the equilibrium statistics in the non-degenerate directions as in Eq. 8. Furthermore, the coefficient of the proportionality relation also depends on $\hat{\nabla}\lambda$. Therefore to test this prediction for how learning rate affects the trace evolution, we train a model multiple times with different $\eta$s. When doing this, the initialization for each run is in equilibrium in the non-degenerate space, and we choose a loss function for which $\hat{\nabla}\lambda$ is constant. We design a simple four-dimensional toy model with the loss $L(\boldsymbol{\theta}) = |\theta_3 + \theta_4|\theta_1^2 + |\theta_3 + 2\theta_4|\theta_2^2$ to test this effect. We initialize the model to make sure that $|\theta_3 + \theta_4|$ and $|\theta_3 + 2\theta_4|$ don't change sign during a few iterations of training. For each $\eta$, we first train for 1000 iterations with SGD noise to equilibrate the model and calculate the trace. Then the model is trained for another 1000 iterations and we calculate the trace evolution. The result is shown in Fig 2(b), which shows a linear relation as predicted.

Next, we measure the evolution of the Hessian trace and determinant after introducing noise with different covariance structure. Our theory predicts that SGD noise decreases the trace while isotropic noise in the non-degenerate subspace decreases the determinant. To test this, we design a toy model where the behavior of these two quantities is anti-correlated. The loss can be written as $L(\boldsymbol{\theta}) = (4 - 2e^{-\theta_3^2 - \theta_4^2})\theta_1^2 + e^{-(\theta_3 - \theta_4)^2}\theta_2^2$. We train the model with the same initialization and with the two different noise structures. The result is shown in Fig. 2(c)-(d), consistent with our analysis.

## 4.3 Trace Evolution along Projected Path

In Section 2, we presented two experiments to support the existence of a minimal valley. Recall that the projected paths follow along the bottom of the valley associated with the SGD dynamics. Our theory predicts that the Hessian trace along the projected paths should decrease during optimization. To confirm this, we use method in Bai et al. (1996) to estimate the Hessian trace and the results are shown in Figure 3, in which the values and error bars are calculated by the mean and standard deviation of 10 different initializations. As seen in the figure, the Hessian trace indeed decreases on average in both cases, which is consistent with our theoretical prediction. Notice that test performance does not monotonically improve during optimization. Indeed the trace continues decreasing slightly even when validation loss goes up, indicating that the trace itself is not sufficient to describe generalization.

## 5 Conclusion

In this paper, we showed that within a minimal valley, the trace of the loss function Hessian decreases on average. We also demonstrated how to design the covariance structure of added noise during optimization to affect other properties of the Hessian's evolution, opening the door to new algorithms. The analysis in this paper can be extended and generalized to models trained on much larger data sets.

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

## A  APPENDIX

### A.1  PROOF OF LEMMA 1

To study the dynamics of SGD, we investigate the model behavior starting from $\boldsymbol{\theta}_t$ to $\boldsymbol{\theta}_{t+1}$ by one-step training and assume that $\boldsymbol{\theta}_{t+1}$ is close to $\boldsymbol{\theta}_t$. In this subsection, we derive the property of the loss function in a small neighbourhood, $B(\boldsymbol{\theta}_t)$, of $\boldsymbol{\theta}_t$, which includes $\boldsymbol{\theta}_{t+1}$. We consider the model in a minimal valley with a set $\mathcal{M}$ of minima and define a projection $\mathcal{P}$ that maps any point $\boldsymbol{\theta}$ in $B(\boldsymbol{\theta}_t)$ to the minimum in $\mathcal{M}$ that is closest to $\boldsymbol{\theta}$, i.e.

$$\mathcal{P}(\boldsymbol{\theta}) \equiv \underset{\boldsymbol{\theta^*} \in \mathcal{M}}{\operatorname{argmin}} \|\boldsymbol{\theta} - \boldsymbol{\theta^*}\|_2 \ . \tag{16}$$

Because $\mathcal{M}$ is connected by definition of minimal valley, we assume that $\mathcal{P}$ is also continuous in $B(\boldsymbol{\theta}_t)$. We denote the dimension of whole parameter space and $\mathcal{M}$ by $N$ and $N - n$, respectively. We can then make a quadratic approximation to the loss function in $B(\boldsymbol{\theta}_t)$,

$$L(\boldsymbol{\theta}) = L^* + \frac{1}{2}(\boldsymbol{\theta} - \mathcal{P}(\boldsymbol{\theta}))^{\mathrm{T}} H(\mathcal{P}(\boldsymbol{\theta}))(\boldsymbol{\theta} - \mathcal{P}(\boldsymbol{\theta})) \ , \tag{17}$$

where we expand the loss function at the minimum that is closest to $\boldsymbol{\theta}$, and $H(\mathcal{P}(\boldsymbol{\theta}))$ is the Hessian depending on the position $\mathcal{P}(\boldsymbol{\theta})$ in the valley. $L^*$ is a constant by definition of a minimal valley and will be ignored in the following derivations.

We define a minimal path to be a smooth path in a minimal valley where every point on the path is a minimum. For any path passing through a minimum $\theta^*$, we have the tangent line of the minimal path at $\theta^*$ being in the null space of the Hessian at $\theta^*$ by definition. Precisely, for an arbitrary minimal path $\boldsymbol{\theta^*}(\mu)\colon [0,1] \to \mathcal{M}$, we have

$$H(\boldsymbol{\theta^*}(\mu))\frac{d\boldsymbol{\theta^*}(\mu)}{du} = 0 \ . \tag{18}$$

Next we have the following lemma,

**Lemma 2.** *Let $J(\boldsymbol{\theta})$ be the $N \times N$ Jacobian matrix of function $\mathcal{P}$ at $\boldsymbol{\theta}$ defined as $J_{ij} = \frac{d\mathcal{P}(\boldsymbol{\theta})_i}{d\theta_j}$. We have*

$$H(\mathcal{P}(\boldsymbol{\theta}))J(\boldsymbol{\theta}) = 0 \ . \tag{19}$$

*Proof.* Notice that $\boldsymbol{\theta^*}(\mu) \equiv \mathcal{P}(\boldsymbol{\theta} + \mu\Delta\boldsymbol{\theta})$ forms a minimal path through $\boldsymbol{\theta}$ where $\mu \in [0,1]$ and $\Delta\boldsymbol{\theta}$ is an arbitrary direction in the parameter space. We have

$$\frac{d\boldsymbol{\theta^*}(\mu)}{du}\bigg|_{\mu=0} = J(\boldsymbol{\theta})\Delta\boldsymbol{\theta} \ . \tag{20}$$

By Eq. 18, we have

$$H(\mathcal{P}(\boldsymbol{\theta}))J(\boldsymbol{\theta})\Delta\boldsymbol{\theta} = 0 \ . \tag{21}$$

Since $\Delta\boldsymbol{\theta}$ is arbitrary, we have Eq. 19. $\qquad\square$

Next we consider the diagonalization of the Hessian as

$$H(\mathcal{P}(\boldsymbol{\theta})) = V(\mathcal{P}(\boldsymbol{\theta}))^{\mathrm{T}} S(\mathcal{P}(\boldsymbol{\theta})) V(\mathcal{P}(\boldsymbol{\theta})) \ , \tag{22}$$

where $S(\mathcal{P}(\boldsymbol{\theta})) = diag(\lambda_1(\mathcal{P}(\boldsymbol{\theta})), ..., \lambda_n(\mathcal{P}(\boldsymbol{\theta})), 0, ..., 0)$ and $\lambda_i > 0$ for $1 \leq i \leq n$. As suggested by Gur-Ari et al. (2018), we assume that the eigenspace change is negligible in the small neighbourhood $B(\boldsymbol{\theta}_t)$ and denote constant orthogonal matrix $V \equiv V(\mathcal{P}(\boldsymbol{\theta}_t)) \approx V(\mathcal{P}(\boldsymbol{\theta}))$.

We then perform the coordinate transformation

$$\boldsymbol{\phi} = V\boldsymbol{\theta} \ , \tag{23}$$

and the loss function becomes

$$L(\boldsymbol{\phi}) = \frac{1}{2} \sum_{i=1}^{n} (\phi_i - \tilde{\mathcal{P}}(\boldsymbol{\phi})_i)^2 \lambda_i(\tilde{\mathcal{P}}(\boldsymbol{\phi})) \ , \tag{24}$$

where $\tilde{\mathcal{P}}(\boldsymbol{\phi})$ is the same projection map as $\mathcal{P}(\boldsymbol{\theta})$ but in the coordinate of $\boldsymbol{\phi}$. Notice here that $\tilde{\mathcal{P}}(\boldsymbol{\phi})$ is the projection to the minimum with the least L-2 distance to $\boldsymbol{\phi}$, because the coordinate transformation is orthogonal and distance preserved. We have

$$\tilde{\mathcal{P}}(\boldsymbol{\phi}) = V\mathcal{P}(\boldsymbol{\theta}) \ . \tag{25}$$

We have the following theorem,

**Proposition 3.** $\tilde{\mathcal{P}}(\boldsymbol{\phi})_i$ *is constant for* $i \leq n$.

*Proof.* In $\boldsymbol{\phi}$-coordinate, by Eq. 23 and 25 the Jacobian of $\tilde{\mathcal{P}}(\boldsymbol{\phi})$, $\tilde{J}(\boldsymbol{\phi})_{ij} \equiv \frac{d\tilde{\mathcal{P}}(\boldsymbol{\phi})_i}{d\phi_j}$, is related to $J(\boldsymbol{\theta})$ as

$$\tilde{J}(\boldsymbol{\phi}) = V J(\boldsymbol{\theta}) V^{\mathrm{T}} \ . \tag{26}$$

Therefore Lemma 2 becomes

$$S(\mathcal{P}(\boldsymbol{\theta}))\tilde{J}(\boldsymbol{\phi}) = 0 \ . \tag{27}$$

Notice that $[S(\mathcal{P}(\boldsymbol{\theta}))\tilde{J}(\boldsymbol{\phi})]_{ij} = \sum_{k=1}^{n} \lambda_k \delta_{ik} \tilde{J}(\boldsymbol{\phi})_{kj}$ where $\delta_{ik}$ is the Kronecker delta function, we have

$$[S(\mathcal{P}(\boldsymbol{\theta}))\tilde{J}(\boldsymbol{\phi})]_{ij} = \lambda_i \tilde{J}(\boldsymbol{\phi})_{ij} \ , \ i \leq n \ . \tag{28}$$

From Eq. 27 and $\lambda_i > 0$, we have

$$\tilde{J}(\boldsymbol{\phi})_{ij} = 0 \ , \ i \leq n \ . \tag{29}$$

Notice that $\tilde{J}(\boldsymbol{\phi})_{ij} \equiv \frac{d\tilde{\mathcal{P}}(\boldsymbol{\phi})_i}{d\phi_j} = 0$ for $i \leq n$ and all $j$, we have that $\tilde{\mathcal{P}}(\boldsymbol{\phi})_i$ is constant for $i \leq n$. $\quad\square$

We denote $\tilde{\mathcal{P}}(\boldsymbol{\phi})_i$ as $\phi_i^*$ for $i \leq n$. Therefore we can define a global translation vector $\boldsymbol{\phi}^* = (\phi_1^*, ..., \phi_n^*, 0, ..., 0)^{\mathrm{T}}$ and consider a new set of coordinate $\boldsymbol{\psi} = \boldsymbol{\phi} - \boldsymbol{\phi}^*$ and have

$$L(\boldsymbol{\psi}) = \frac{1}{2} \sum_{i=1}^{n} \psi_i^2 \lambda_i(\bar{\mathcal{P}}(\boldsymbol{\psi})) \ , \tag{30}$$

where $\bar{\mathcal{P}}(\boldsymbol{\psi})$ is also the projection to the minimum with the least L-2 distance to $\boldsymbol{\psi}$.

Clearly we have the set of minima of the loss function 30 as $\{\boldsymbol{\psi}|\psi_i = 0, i \leq n\}$ and the closest minimum to $\boldsymbol{\psi}$ in the L-2 distance is $\boldsymbol{\psi}^* = (0, ..., 0, \psi_{n+1}, ..., \psi_N)^{\mathrm{T}}$. Therefore we have

$$\bar{\mathcal{P}}(\boldsymbol{\psi}) = (0, ..., 0, \psi_{n+1}, ..., \psi_N)^{\mathrm{T}} \ , \tag{31}$$

and

$$L(\boldsymbol{\psi}) = \frac{1}{2} \sum_{i=1}^{n} \psi_i^2 \lambda_i(\psi_{n+1}, ..., \psi_N) \ , \tag{32}$$

Notice that the coordinate transformations from $\boldsymbol{\theta}$ to $\boldsymbol{\psi}$ only involve orthogonal rotation and translation, the SGD rule is conservative. To see this, we first have

$$\nabla_{\boldsymbol{\theta}} = V^{\mathrm{T}} \nabla_{\boldsymbol{\phi}} = V^{\mathrm{T}} \nabla_{\boldsymbol{\psi}} \ . \tag{33}$$

And the SGD on $\boldsymbol{\theta}$ becomes

$$\begin{aligned} \boldsymbol{\theta}_{t+1} &= \boldsymbol{\theta}_t - \eta V^{\mathrm{T}} \nabla_{\boldsymbol{\psi}} L(\boldsymbol{\psi}) \\ \Rightarrow \boldsymbol{\phi}_{t+1} &= \boldsymbol{\phi}_t - \eta \nabla_{\boldsymbol{\psi}} L(\boldsymbol{\psi}) \\ \Rightarrow \boldsymbol{\psi}_{t+1} &= \boldsymbol{\psi}_t - \eta \nabla_{\boldsymbol{\psi}} L(\boldsymbol{\psi}) \ . \end{aligned} \tag{34}$$

In the main text, we replace the notation $\boldsymbol{\psi}$ with $\boldsymbol{\theta}$ for symbol simplicity.

## A.2 Additional Experiments

To further support our assumption of connected minima, we conducted extra experiments including DenseNet (Huang et al., 2017) on CIFAR100, and explored the landscape by introducing randomized labels on a subset of the training data to initialize subsequent training on purely clean data (details in Section A.2.2).

### A.2.1 DenseNet on CIFAR100

We trained a DenseNet with batch normalization (number of blocks = [6,12,24,16] and growth rate=12) on CIFAR100 and followed the same pipeline as described in the main text experiments. The learning rate is $0.1$ and the batch size is $512$. The result is shown in Figure 4(a) which supports the connectedness assumption as well as the result of the Hessian trace decreasing under SGD.

### A.2.2 4-layer ConvNet on MNIST and ResNet18 on CIFAR10 with Randomized Labels

The setup of these experiments is as follows: we randomly split the training set into two sets, A and B. For MNIST and CIFAR10 we let the number of samples in A be $40000$. Then we shuffle the labels in set B. For the rest of this section, we call A the training set and B the shuffle set. The loss minimum, Hessian trace, etc. are all with respective to the training set A.

First we will obtain minima with worse generalization than the minima found by regular SGD. To do so, we train the model with the union of the training set A and shuffle set B until convergence. Then we further train the model with only A using gradient descent so the model reaches a minimum. This minimum corresponds to the left end point of the path in Figure 4 (b) and (c).

We then train the model from this minimum with training set A alone using SGD. We find that the model starts to diffuse and validation loss becomes smaller even though it is at minimum already. Using the same method described in main text, we project this path to minimal and the results are shown in Figure 4 (b) and (c).

In both cases, we found that minima found by SGD are connected, and we further estimated the trace along both paths and found that it is decreasing on average. These results support the connectedness and degeneracy assumption and our result of SGD decreasing Hessian on average. It also illustrates how SGD helps model explore the loss landscape and escape bad minima.

## A.3 The Sum of Negative Eigenvalues

In fact, if we consider an equilibrium of $\theta_i$ for $i \leq n$, we have $\langle L \rangle = L^* + C \sum_{i=1}^{n} \lambda_i$ and $\langle \partial_j \partial_k L \rangle = C \sum_{i=1}^{n} \partial_j \partial_k \lambda_i$ for $j, k > n$ from Eq. 11, where $C = \frac{n}{4S}(1 - \frac{S}{M})$. Thus as we change learning rate or batch size, and hence $C$, $\langle \partial_j \partial_k L \rangle \propto \langle L \rangle - L^*$ for states nearby in the minimal valley because both are proportional to $C$ and $\sum_{i=1}^{n} \lambda_i$ and $\partial_j \partial_k \lambda_i$ are approximately constant for states nearby in the minimal valley. Usually in deep neural networks training loss can go to nearly zero with a large number of training iterations and small learning rate (Zhang et al., 2017; Du et al., 2019), so we ignore $L^*$ and have that $\langle \partial_j \partial_k L \rangle \propto \langle L \rangle$. This result is tested and verified in Figure 2(a).

## A.4 Experiments on the Hessian-Noise Covariance Alignment Assumption

To further support our assumption of the Hessian-noise covariance alignment, we trained 2-layer neural networks on the CIFAR10 dataset with label-smoothed cross-entropy loss and mean-squared loss (MSE) and calculated the cosine similarity between the Hessian of the loss ($H$) and the noise covariance ($C$).

The cosine similarity between two matrices $A$ and $B$ is defined as

$$CS(A, B) = \frac{\text{Tr}(AB^T)}{\sqrt{\text{Tr}(AA^T)\text{Tr}(BB^T)}} \ , \tag{35}$$

where Tr is the trace operator. It is easy to show that $A$ and $B$ are aligned as $A = kB, \ k > 0$ if and only if $CS(A, B) = 1$.

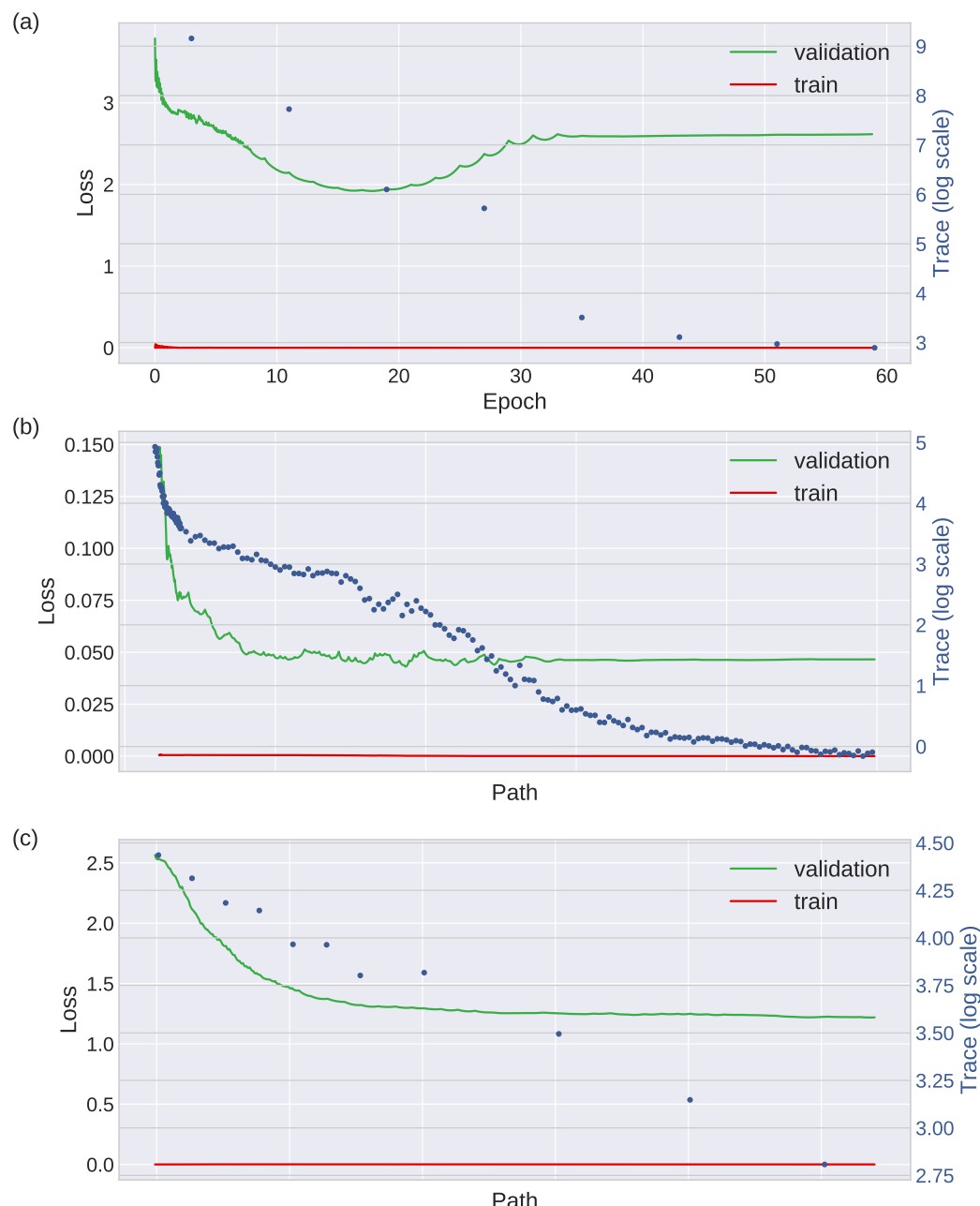

Figure 4: (a) Experiments performed as in the main text with DenseNet on CIFAR100. (b) Experiments performed as described in Section 1.2 using 4-layer ConvNet on MNIST and (c) Preact-ResNet18 on CIFAR10.

In order to make calculate of the Hessian and covariance tractable, the networks were chosen to have one hidden layer with 70 nodes, and the dataset is transformed into grey scale and resized to $7 \times 7$ pixels and normalized. The network with label-smoothed cross-entropy loss is trained with SGD optimization with learning rate $0.001$ and batch size $512$ and the network with mean-squared loss is trained with SGD optimization with learning rate $0.1$ and batch size $512$.

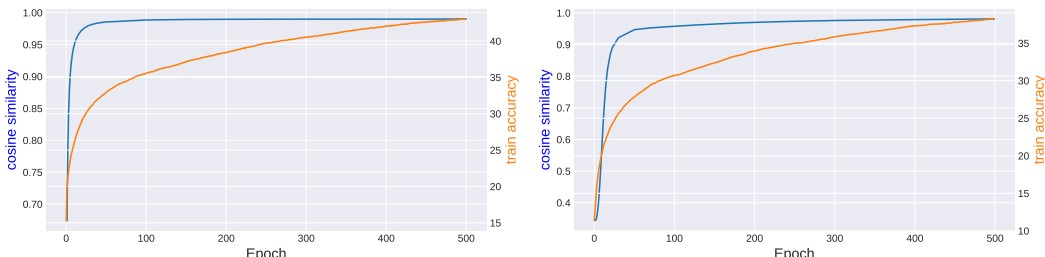

Figure 5: The cosine similarity between $H$ and $C$ and training curve of 2-layer networks trained on CIFAR10 with label-smoothing cross-entropy loss (left) and mean-squared loss (right).

As shown in Figure 5, in both cases the cosine similarity between the Hessian and noise covariance is larger than $0.9$ after several epochs of initial training. Notice that two random matrices with the same size as $H$ and $C$ in these experiments are almost orthogonal with the cosine similarity of $0$. These experiments are consistent with our assumption of Hessian-noise covariance alignment–after the early phase of training–in the main text.

### A.5 EXPERIMENT ON THE TIMESCALE SEPARATION ASSUMPTION

In this section, we provide support for our assumption of timescale separation through experiments with a small convolutional neural network trained on CIFAR10. In order to observe the relaxation to steady-state, we adopt the method in Yaida (2019), which shows that

$$\langle \boldsymbol{\theta} \cdot \nabla L \rangle = \frac{\eta(1+\mu)}{2(1-\nu)} \langle \boldsymbol{v}^2 \rangle \tag{FDR}$$

is satisfied when the model is at stationarity, where $\mu$ and $\nu$ are the momentum and dampening in SGD optimization, respectively, and $v$ is the velocity. It is easy to show that

$$\langle \mathcal{P}(\boldsymbol{\theta}) \cdot \mathcal{P}(\nabla L) \rangle = \frac{\eta(1+\mu)}{2(1-\nu)} \langle \mathcal{P}(\boldsymbol{v})^2 \rangle \;, \tag{FDR'}$$

when there exists a stationary state in a subspace $\mathbb{S}$ and $\mathcal{P}$ is the projection operator to $\mathbb{S}$. We use $\mathcal{O}_L$ and $\mathcal{O}_R$ to denote the quantities of the left and right side respectively of FDR and FDR'. Therefore the quantity $|\bar{\mathcal{O}}_L/\bar{\mathcal{O}}_R - 1|$ can be used as a criterion to verify how close the model is to a stationary state in a specific subspace.

To show the existence of timescale separation, we train a small CNN on CIFAR10. This network has 3 convolutional layers with kernel size of 3 and output channel number of 15, 20, 10 respectively. Throughout training, we estimate $|\bar{\mathcal{O}}_L/\bar{\mathcal{O}}_R - 1|$ on the raw parameter space and top-1 eigenspace of the Hessian. The result is shown in Figure 6.

Figure 6 shows that the criterion in top-1 eigenvalue decays much faster than in the raw parameter space. After several epochs, $|\bar{\mathcal{O}}_L/\bar{\mathcal{O}}_R - 1|$ in top-1 subspace is already less then $10^{-2}$ which is a threshold set in Yaida (2019) to indicate sufficient closeness to a stationary state, while the state in the whole parameter space is still far from equilibrium after 50 epoch of training. This experiment fully supports our assumption of timescale separation in the main text.

### A.6 EXTENSION TO NON-DEGENERATE LANDSCAPE

This section explores the situation where the minimal valley assumption is slightly violated, in which case the bottom of the landscape is not completely degenerate but has a small slope. The loss function 1 can then be written as

$$L(\boldsymbol{\theta}) = L^* + \frac{1}{2} \sum_{i=1}^{n} \theta_i^2 \lambda_i(\theta_{n+1}, ..., \theta_N) + g(\theta_{n+1}, ..., \theta_N) \;, \tag{36}$$

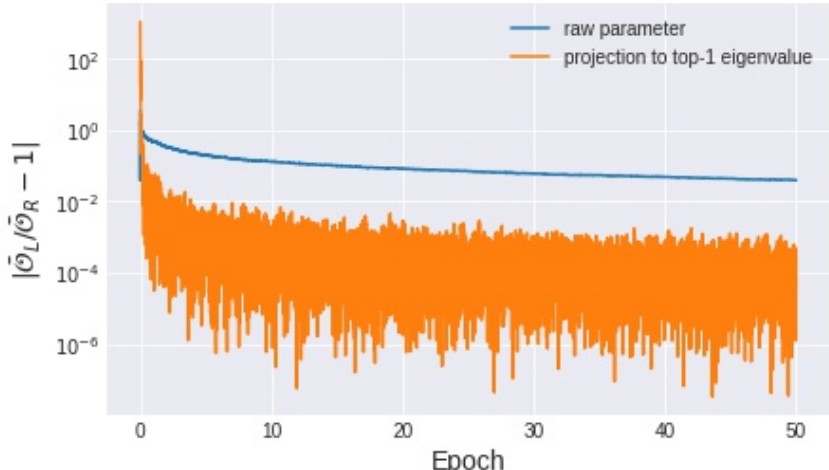

Figure 6: $|\bar{\mathcal{O}}_L/\bar{\mathcal{O}}_R - 1|$ plots along training. The smaller this quantity is, the closer a state is to a equilibrium. This shows that a state in the top-1 eigenspace of the Hessian decays much faster to equilibrium then in the raw parameter space and supports the assumption of timescale separation.

where the function $g$ describes how the bottom of the valley varies in the previously degenerate directions. Following the same proof as in Section 2.2, it is easy to obtain

$$\langle [\![T_{t+1} - T_t]\!]_{\mathrm{m.b.}} \rangle \approx -\frac{\eta^2}{4S}\left(1 - \frac{S}{M}\right)\left\|\sum_{i=1}^{n}\hat{\nabla}\lambda_i(\hat{\boldsymbol{\theta}})\right\|^2 - \eta\sum_{i=1}^{n}\hat{\nabla}\lambda_i(\hat{\boldsymbol{\theta}})\hat{\nabla}g(\hat{\boldsymbol{\theta}}) \ . \tag{37}$$

At the end of training, the model reaches equilibrium, where

$$\langle [\![T_{t+1} - T_t]\!]_{\mathrm{m.b.}} \rangle = 0 \ .$$

Thus, from Equation 37 we have

$$\hat{\nabla}g(\hat{\boldsymbol{\theta}})\sum_{i=1}^{n}\hat{\nabla}\lambda_i(\hat{\boldsymbol{\theta}}) = -\frac{\eta}{4S}\left(1 - \frac{S}{M}\right)\left\|\sum_{i=1}^{n}\hat{\nabla}\lambda_i(\hat{\boldsymbol{\theta}})\right\|^2 \tag{38}$$

This reveals explicitly the effect of SGD noise as an entropic force during training. The training of gradient descent and its variant drives the model to states with $\bar{\boldsymbol{\theta}} = 0$ and $\hat{\nabla}g(\hat{\boldsymbol{\theta}}) = 0$ according to the energy function 36. This exert energetic force to the model moving forward to the lowest energy (smallest loss). However when training with noise, $\hat{\nabla}g(\hat{\boldsymbol{\theta}}) = 0$ may not be satisfied when the locations with smallest trace and $\hat{\nabla}g(\hat{\boldsymbol{\theta}}) = 0$ do not coincide because the right hand side of Equation 38 is always greater or equal to zero. That being said the model is pushed away from $\hat{\nabla}g(\hat{\boldsymbol{\theta}}) = 0$ as a entropic competitor to the energetic force.

We use a toy model to visualize this effect. Consider a loss function of $L(\theta_1, \theta_2) = \frac{1}{2}(\theta_2^2+1)\theta_1^2+\alpha*\theta_2$. We train this function by GD and SGD with $\alpha = 10^{-4}$, learning rate $\eta = 0.01$, noise strength $\frac{1}{S} - \frac{1}{M} = 1$ and initialization $(\theta_1, \theta_2) = (0.1, 0.1)$. The result is shown in Figure 7. The left column corresponds to training with gradient descent where $\theta_1$ drops to zero very fast and $\theta_2$ continues decreasing to negative infinity. As when noise is introduced, $\theta_2$ stops decrease after equilibrium between entropic and energetic force. From 38 it is easy to show that $\langle\theta_2\rangle = -0.02$, which is consistent with the right column of Figure 7.

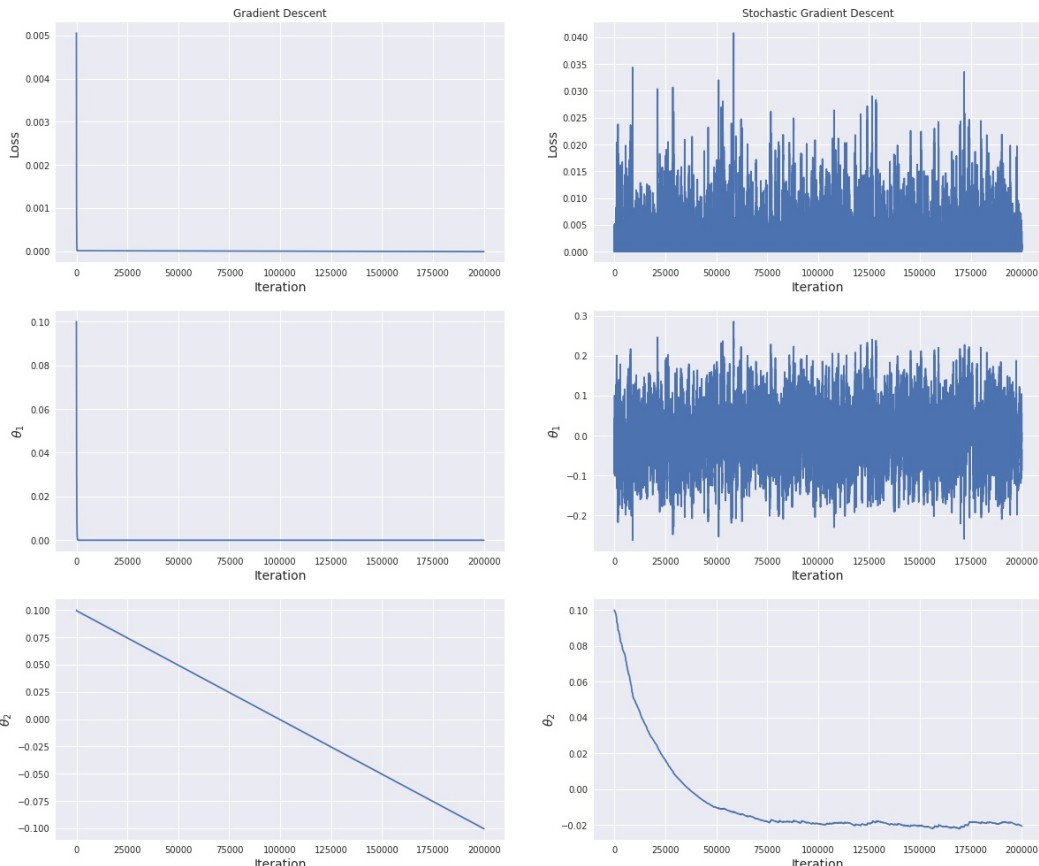

Figure 7: Toy model $L(\theta_1, \theta_2) = \frac{1}{2}(\theta_2^2 + 1)\theta_1^2 + \alpha * \theta_2$ trained with gradient descent (left) and noisy gradient descent (right).

