# OpenReview forum: "How noise affects the Hessian spectrum in overparameterized neural networks"
_ICLR.cc/2020/Conference — Reject_

### Official Review · AnonReviewer3 · 2019-10-22
**Official Blind Review #3**

**Rating:** 6

**Review:**

This paper considers the problem of how noise coming from the gradient-based update affects the geometry of the hessian matrix when training a neural network.

The paper makes an interesting claim that around a local minimum, if the noise in SGD is aligned with the hessian matrix of the network, then doing SGD update is implicitly minimizing the trace of the hessian matrix, biasing the current point towards a wide valley.

The paper also makes a very interesting observation that isotropic noise will decrease the determinate of the hessian while the SGD noise will decrease the trace of the hessian matrix.

I find the theorem in the paper quite interesting, especially Lemma 1 stating that the loss function can be locally approximated by a quadratic function whose variables are the non-degenerate directions and the coefficients only depends on the degenerate directions. This Lemma appears to be quite novel to me. With this Lemma, it is then easy to see that as long as the noise is aligned with the non-degenerate directions, then the trace of the Hessian is decreasing in expectation at every step.

The main question I have about this paper is the clarity: First of all, the main concept "noise covariance matrix aligned with the hessian" is not mathematically defined anywhere. I can intuitively understand the term from the explanation in section 2.1.2 but I can not formally justify the correctness. Second, where is the timescale separation used? Is it for justifying the assumption of the local stationary point approximation or for Lemma 1 or something else? At the current level of writing in the paper, I can not formally verity the theorems.

Missing citation: "an alternative view, when does SGD escape local minimal."



After Rebuttal: I have read the authors' responses and acknowledge the sensibility of the statement.





**Experience Assessment:**

I have published in this field for several years.

**Review Assessment: Checking Correctness Of Derivations And Theory:**

I assessed the sensibility of the derivations and theory.

**Review Assessment: Checking Correctness Of Experiments:**

I did not assess the experiments.

**Review Assessment: Thoroughness In Paper Reading:**

I read the paper at least twice and used my best judgement in assessing the paper.

---

> ### Author Response · Authors · 2019-11-13
> **Response to Reviewer #3**
>
> We thank the reviewer for the detailed review and questions. We hope the following answer your questions.
>
> 1. In the updated draft, we measure alignment of the noise covariance and Hessian using equation (35) and conducted extra experiments to support the alignment assumption. These results are included in A.4 of the updated version. We trained two networks (using either label smoothed cross entropy or squared error) on CIFAR10 and calculated H and C throughout training. The results show that the cosine similarity between the Hessian and noise covariance--which is 1 if and only if two matrices are the same up to a constant positive factor--is greater than $0.9$ after a few epochs of initial training. Notice that two random matrices with the same size as $H$ and $C$ in these experiments are nearly orthogonal with the cosine similarity having mean zero and std of 1/sqrt{number of parameters}. These experiments are consistent with our assumption of Hessian-noise covariance alignment in the main text. Full details can be found in the appendix.
>
> 2. Based on Lemma 1, we can separate the reparameterized parameters into a non-degenerate set (\bar{\theta}) and a degenerate set (\hat{\theta}), which correspond to non-zero and zero (or extremely small) Hessian eigenvalues, respectively. The timescale separation assumes that relaxation to the stationary state in the non-degenerate directions is fast compared to motion in the degenerate directions. In physics/thermodynamics, this would be a quasi-static process. The slow evolution of the degenerate parameters results in the decrease of Hessian trace.
>
> A similar question was also asked by Reviewer 1 and we restate our response to that here: To provide experimental evidence justifying the timescale separation assumption, we performed an extra experiment on a CNN trained on CIFAR10, presented in A.5 of the updated draft. For this experiment, we implement the criterion from Yaida, (2019) to measure the proximity to a steady-state and found that the top-1 eigenspace meets the criterion after only 5 epochs of training, while there is no steady-state in the whole model parameter space after 50 epochs of training. This supports the existence of a timescale separation.
>
> Sho Yaida, 2019, Fluctuation-dissipation relations for stochastic gradient descent.

---

### Official Review · AnonReviewer1 · 2019-10-22
**Official Blind Review #1**

**Rating:** 3

**Review:**

This paper proves that under certain conditions, SGD on average decreases the
trace of the Hessian of the loss. The paper is overall clear and the topic addressed in the paper is relevant in machine learning but I don’t think this paper is ready for publication. There are numerous assumptions that are made in the paper, but not properly stated or backed up. In my view, the experimental results are also not sufficient to compensate for the shortcomings of the theoretical part.

Lemma 1
This is a corollary of Morse lemma, which is also used in Dauphin et al: https://arxiv.org/pdf/1406.2572.pdf (see Equation 1). I don’t see why you would need to re-derive it in your paper and most importantly this should be clearly stated in your paper.

Page 2: existence stationary state?
What are the conditions for the existence of a stationary state? Is bounded noise sufficient?

Equation 4 and local approximation
Regarding the standard decomposition of the Hessian in eq 4 (sometimes referred to as Gauss-Newton decomposition), the discussion is not precise. The authors claim “Empirically, negative eigenvalues are exponentially small in magnitude compared to major positive ones when a model is close to a minimum“
But clearly from the formula itself, one needs to differentiate between minima with low and high function values. For local minima that are high in the energy landscape, the second term might have a large function value. If of course all **depends on the size of the approximation region**. This is extremely important and it is not properly characterized in the paper. The authors simply claim that the loss can be locally approximated while I think these assumptions should be clearly stated. Note that similar types of analyses are usually done in dynamical systems where the behavior of a system is linearized around a critical point. In some cases, one can however characterize the size of a basin of attraction, see e.g. the book Nonlinear Systems (3rd Edition): Hassan K. Khalil.

Assumption on timescale separation
This section makes numerous claims that are not properly backed up.
1) “This assumption is because there is a timescale separation between
the dynamics of \theta_bar, which relax quickly and the dynamics of \theta_hat, which evolve much more slowly as the minimal valley is traversed.“
Are you claiming the absolute value of the eigenvalue of the non-degenerate space are much smaller than the degenerate space? Why would that be so?
2) Ornstein-Uhlenbeck: OU processes require the noise to be Brownian motion. This assumption needs to be clearly stated. Note that there is actually evidence that the noise of SGD is heavy-tail:
Simsekli, Umut, Levent Sagun, and Mert Gurbuzbalaban. "A tail-index analysis of stochastic gradient noise in deep neural networks." arXiv preprint arXiv:1901.06053 (2019).

Take away
I am unsure what the added value of this paper is. Second-order methods can already be shown to decrease the maximum eigenvalue, see e.g. convergence results derived in Cubic regularization of Newton method and its global performance by Nesterov and Polyak. These methods have also been analyzed in stochastic settings where similar convergence results hold as well. What particular insight do we gain from the results of Theorem 2? The authors claim this could “potentially improve generalization” but this is not justified and no reference is cited.

“This indicates that the trace itself is not sufficient to describe generalization (Neyshabur et al., 2017).“
I do not see what aspect of Neyshabur et al. justifies your claim, please explain.

Experiments
All the experiments performed in the paper are on very small models. Given the rather strong assumptions made in the paper, I feel that the paper should provide stronger empirical evidence to back up their claims.


**Experience Assessment:**

I have published in this field for several years.

**Review Assessment: Checking Correctness Of Derivations And Theory:**

I carefully checked the derivations and theory.

**Review Assessment: Checking Correctness Of Experiments:**

I assessed the sensibility of the experiments.

**Review Assessment: Thoroughness In Paper Reading:**

I read the paper thoroughly.

---

> ### Author Response · Authors · 2019-11-13
> **Response to Reviewer #1**
>
> We thank the reviewer for the detailed review and questions. We hope the following address your concerns.
>
> 1. Morse’s lemma applies to expansion over a point, whereby the curvature is a constant. In our work, we allow the curvature to be a function of the degenerate directions, which is critical to our work. One can view our lemma as a generalization of Morse lemma that incorporates this. We have updated our draft and state this more clearly in the new version.
>
> 2. This question was also asked by Reviewer 2, and we restate our response here:
> Experiments show that relaxation to a steady state is typical in neural network training [Figure 1 in Yaida (2019), Figure 6 in our paper] and support timescale separation [Figure 6 in our paper]. A nontrivial case where no equilibrium exists, as was mentioned in the review, is models trained with cross entropy loss without a regularizer, where model parameters logarithmically diverge and never reach stationarity. This is because there is no local minimum with finite model parameters. However, in common practice, where neural networks are trained with regularization (such as L^2 regularization or label-smoothed cross entropy), this can be avoided, as also discussed in 2.3.4 of Yaida (2019). We will update the discussion in the draft to reflect this point.
>
> 3. We understand this question to be related to our assumption that the noise covariance matrix is aligned with the Hessian. Please correct us if we misunderstood. We agree that the evidence we provided for this assumption was lacking, so we performed new experiments, presented in A.4 of the appendix. This question was also asked by Reviewer 3 and we reproduce our response to them here:
>
> To test this assumption, we trained two neural networks on CIFAR10 and calculated H and C throughout training. The results show that the cosine similarity between the Hessian and noise covariance--which is 1 if and only if two matrices are the same up to a constant positive factor--is greater than $0.9$ after a few epochs of initial training. Notice that two random matrices with the same size as $H$ and $C$ in these experiments are nearly orthogonal with the cosine similarity having mean zero and std of 1/sqrt(number of parameter). These experiments are consistent with our assumption of Hessian-noise covariance alignment in the main text. Full details can be found in the appendix.
>
> 4. We use Ornstein-Uhlenbeck to provide intuition on the occurrence of timescale separation. This can be generalized to Lévy-driven Ornstein–Uhlenbeck processes, which may describe SGD [Simsekli et al., 2019]. As shown in page 1033 and 1034 of [Abdelrazeq et al. 2014], the decay rate is also proportional to the eigenvalues of the Hessian with quadratic loss. We have updated this statement in the latest version.
>
> We decided to go further and run experiments instead of relying on the above argument. To provide experimental evidence justifying the timescale separation assumption, we performed an experiment on a CNN trained on CIFAR10, presented in A.5 of the updated draft. For this experiment, we implemented the criterion from Yaida, (2019) to measure the proximity to a steady-state and found that the top-1 eigenspace meets the criterion after only 5 epochs of training, while there is no steady-state in the whole model parameter space after 50 epochs of training. This supports the existence of a timescale separation.
>
> 5. Our theorem 2 applies to any monotonic function of the eigenvalues of the Hessian by introducing noise into first-order training. Decreasing the maximum eigenvalue can be seen as a special case of our result. Theorem 1 also explains how a first-order method, SGD, decreases the trace of the Hessian.
>
> We apologize for the confusing citation here, and we have removed it in the latest version. The point of this statement, as also emphasized by Reviewer 2, was that the trace does not seem to predict generalization. This fact motivates our analysis of other added noise structures that can affect other properties of the Hessian spectrum.
>
> 6. We use VGG16, PreAct-ResNet18 and DenseNet trained on CIFAR10 and CIFAR100 to verify the decrease of the Hessian trace, and extra experiments on fully-connected and convolutional networks to support our assumptions. We believe that carefully constructed experiments on these models and datasets make a strong case for the validity of our assumptions and results.
>
> Sho Yaida, 2019, Fluctuation-dissipation relations for stochastic gradient descent.
> Abdelrazeq et al., 2014, Model veriﬁcation for levy-driven ornstein-uhlenbeck processes
> Simsekli et al., 2019, A tail-index analysis of stochastic gradient noise in deep neural networks.

---

> > ### Comment · AnonReviewer1 · 2019-11-13
> > **Clarification needed regarding claims about generalization**
> >
> > I would like to thank the authors for clarifying some of my concerns. However, I still have two important concerns that have not been addressed.
> >
> > 1. Additional experiments computing the cosine similarity between H and C. You are optimizing a 2-layer neural network for 500 epochs but it seems it converges after 20 epochs. before convergence, the similarity starts at 70%, there is therefore a significant share of the Hessian that does not come from the covariance. I'm therefore not convinced this is a valid assumption. I would still expect this point to be clarified in the paper as I believe this is still a strong assumption to make.
> >
> > 2. This is to me more important than my first comment as I still fail to see the contribution made by the Theorems derived in the paper. The authors claimed in their paper that their results relate to generalization but they acknowledge that "the trace does not seem to predict generalization.". This seems to be two contradictory statements to me.

---

> > > ### Author Response · Authors · 2019-11-13
> > > **Response to Reviewer #1**
> > >
> > > Thank you very much for the rapid response. We believe we can clarify both points:
> > >
> > > 1. We did not make clear enough, and will do so in a revised version, that we only claim our assumptions hold after an initial phase of training, corresponding to ~10 epochs in our main text experiments. During this early phase, the local loss landscape is not well described by a “minimal valley” picture, nor is the noise aligned with the Hessian. The distinct nature of the early phase of training has been recently studied [1]. Also note that at initialization the Hessian has large negative eigenvalues. However, as observed in Figure 2 of [Ghorbani et al., 2019], these disappear rapidly after about 400 iterations. We claim our assumptions hold after this early phase is complete and will make this clear in an updated draft.
> > >
> > > 2. We did not intend to claim that our results speak directly to generalization. We study what noise does to the Hessian, both SGD and added noise with chosen covariance. How the Hessian relates to generalization is an important open question. While generalization bounds related to norm-based measures have been popular in the past few years, recent experiments, in particular section 8 of [3], indicate that flatness measures have the best causal relationship to generalization. With our generalized theorem about added noise, when future work makes progress on proving flatness based generalization bounds, we can add noise that will push on the relevant quantity. This is an exciting prospect, but we believe that proving such generalization bounds is a different project. We will modify the paper to ensure we do not overclaim what we have done.
> > >
> > > [1] https://openreview.net/forum?id=Hkl1iRNFwS
> > > [2] Ghorbani et al., 2019, An Investigation into Neural Net Optimization via Hessian Eigenvalue Density.
> > > [3] https://openreview.net/forum?id=SJgIPJBFvH

---

### Official Review · AnonReviewer2 · 2019-10-29
**Official Blind Review #2**

**Rating:** 6

**Review:**

This paper studies the behavior of SGD after it has reached a steady state and settled in a minimal valley. The authors show that even if SGD has reached a minimal valley, the noisy updates provided by a minibatch of samples result in the trace of the Hessians reducing with further SGD iterations. The authors also show that by changing the type of noise that is added to GD, one can get different functions of the Hessian to reduce with SGD iterations. The theoretical conclusions are then verified empirically with synthetic examples and real deep learning examples

In my opinion this is an interesting paper and research direction that helps us understand how SGD biases solutions towards "flatter" minima as measured by the trace norm of the Hessian of the loss function. I have a few concerns though:

1. Is the steady state assumption valid for neural network training? Especially when training on exponential losses (like cross entropy) which drives the parameters towards large norm solutions? This seems to be an important yet unsupported assumption in the analysis.

2. Does the Hessian-noise covariance alignment exist for squared loss functions as well? Is this specific to log-likelihood models?

3. In the experiments to delineate how different types of noise can result in regularization of different quantities, is there a reason only a synthetic example is used? Can this be replicated for deep networks, or are the two quantities (trace vs determinant) closely related? It would be nice to see how this behavior extends to deep networks.

Additional questions/comments:
1. While overparameterization seems to be important to the analysis (the parameters move in the degenerate subspace but not in the non-degenerate one), is there anyway one can amend this framework to analyze different levels of overparameterization? Is there any difference in the rates of decrease in the trace norms for highly overparameterized models vs lightly overparameterized ones?

2. This paper talks about how SGD has a bias towards flatter minima. Is there a reason to prefer flatter solutions over sharper ones to get better generalization? Can one prove this connection?

3. In the Densenet experiments in Figure 4a, it seems as though this relationship between flatness and generalization might not hold? There are points along the optimization trajectory where the validation loss seems to be better than at the last few points along the optimization path. However at the former set of points the trace norm of the hessian is larger than at the latter points. Is this from a stray experiment, or are the plots averaged over a number of different runs? Please add these details as well as other details such as learning rates, criteria to decide when steady state was reached, whether or not batch norm was used, etc.

4. The Fluctuation-Dissipation Relations need to be explained more clearly. For people unfamiliar with statistical physics (and Yaida 2019) the notation and formulation is not immediately clear and that makes the paper harder to read.

Overall I believe this paper explains an important phenomenon. I am willing to update my score if my concerns are addressed/if there is any misunderstanding.

**Experience Assessment:**

I have read many papers in this area.

**Review Assessment: Checking Correctness Of Derivations And Theory:**

I assessed the sensibility of the derivations and theory.

**Review Assessment: Checking Correctness Of Experiments:**

I assessed the sensibility of the experiments.

**Review Assessment: Thoroughness In Paper Reading:**

I read the paper thoroughly.

---

> ### Author Response · Authors · 2019-11-13
> **Response to Reviewer #2**
>
> We thank the reviewer for the detailed review and questions. We hope the following address your concerns.
>
> 1. Experiments show that relaxation to a steady state is typical in neural network training [Figure 1 in Yaida, 2019, Figure 6 in our paper] and support timescale separation [Figure 6 in our paper]. A nontrivial case where no equilibrium exists, as was mentioned in the review, is models trained with cross entropy loss without a regularizer, where model parameters logarithmically diverge and never reach stationarity. This is because there is no local minimum with finite model parameters. However, in common practice, where neural networks are trained with regularization (such as L^2 regularization or label-smoothed cross entropy), this can be avoided, as is also discussed in 2.3.4 of Yaida (2019). We will update the discussion in the draft to reflect this point.
>
> 2. Yes, our experiments in Figure 5 of the Appendix of the updated draft show the Hessian-noise covariance alignment for both cross-entropy and mean squared loss. Note of course that mean squared loss can also be viewed as a log-likelihood model as L(y, y’) = -log Aexp(-(y-y’)^2) with Gaussian prior.
>
> 3. The quantities seem to be closely related in many of the network architectures we studied. We used the synthetic example to validate our analysis in a setting where the difference in behavior is very clear. It would be interesting to explore network structures that have dramatically different behavior with respect to trace, determinant, maximal eigenvalue, etc.
>
> For additional questions,
> 1. This is a good question. While it’s difficult to formalize the relationship between overparameterization and degeneracy, intuitively the degree of overparameterization positively affects the degree of loss landscape degeneracy. To make some progress, in A.6 of our updated draft, we amend this framework to analyze models with less overparameterization by imaging the flat directions being slightly tilted. We show that there can be a competition between the “entropic” force due to the noise and an “energetic” force due to the gradient, in particular when the directions with decreasing trace and decreasing loss are at odds. These two effects will equilibrate following Equation 38 in the updated draft.
>
> 2. The relationship between flat minima and generalization is an active area of research. [Hochreiter & Schmidhuber, 1997; Keskar et al., 2017; Dziugaite & Roy, 2017; Smith & Le, 2018] provides support for the idea of flatness leading to good generalization, PAC-Bayesian theory, etc while [Dinh et al., 2017] questions this relation. Even the definition of flatness is controversial as it depends on the choice of parameterization. In this paper, we focus strictly on what noise does to the Hessian spectrum, rather than what properties would be desirable. Notably, we have a general result on how different properties of the spectrum can be affected by proper choice of added noise, which will be useful once the community has a better handle on how flatness relates to generalization.
>
> 3. We have updated the draft to include all necessary hyperparameters. This is a relatively small DenseNet on CIFAR100 and therefore the effect of overfitting is significant which is why the validation loss rises. Note that we observe that the trace is decreasing regardless of the test performance. As discussed above, we do not know the relationship between flatness and generalization, and therefore the model can and does overfit even while the trace decreases. This fact indeed motivates our generalization to other added noise structures that may ultimately provide superior generalization performance to SGD noise.
>
> 4. We agree and will modify our draft and include a more detailed introduction and exposition in the next version of draft.
>
> Sho Yaida, 2019, Fluctuation-dissipation relations for stochastic gradient descent.
> S. Hochreiter and J. Schmidhuber. 1997, Flat minima.
> Keskar et al., 2017, On large-batch training for deep learning: Generalization gap and sharp minima.
> Dziugaite & Roy, 2017, Computing nonvacuous generalization bounds for deep (stochastic) neural networks with many more parameters than training data.
> Smith & Le, 2018, A bayesian perspective on generalization and stochastic gradient descent.
> Dinh et al., 2017, Sharp Minima Can Generalize For Deep Nets

---

### Author Response · Authors · 2019-11-13
**General response to all reviewers**

We thank all reviewers for their valuable feedback and helpful comments. To address the questions raised, we made the following major modifications to our paper:

1. We have added an extra experiment with models trained on CIFAR10 to support our assumption of Hessian-noise covariance alignment. The results are presented in A.4.
2. Additional experiments were performed and added to A.5 to support the timescale separation assumption. We also modified the analysis in 2.1.3.
3. We have added additional analysis and results for less overparameterized models in the case where the degenerate directions have a gradient, and show that noise can prevent the model from moving down the gradient.
4. We have clarified our claims in the latest version, made more clear the assumptions we make, and added the missing citations as appropriate.

In addition, a number of smaller modifications to the presentation have been made, following the useful feedback provided. Overall, we believe these modifications to both the content and presentation have dramatically improved the work. Once again, we thank the reviewers for their efforts which we believe have greatly improved the paper.

---

### Decision · Program_Chairs · 2019-12-19

**Decision:**

Reject

**Comment:**

The study of the impact of the noise on the Hessian is interesting and I commend the authors for attacking this difficult problem. After the rebuttal and discussion, the reviewers had two concerns:
- The strength of the assumptions of the theorem
- Assuming the assumptions are reasonable, the conclusions to draw given the current weak link between Hessian and generalization.

I'm confident the authors will be able to address these issues for a later submission.